# Artificial Intelligence in Lung Cancer Imaging: Unfolding the Future

**DOI:** 10.3390/diagnostics12112644

**Published:** 2022-10-31

**Authors:** Michaela Cellina, Maurizio Cè, Giovanni Irmici, Velio Ascenti, Natallia Khenkina, Marco Toto-Brocchi, Carlo Martinenghi, Sergio Papa, Gianpaolo Carrafiello

**Affiliations:** 1Radiology Department, Fatebenefratelli Hospital, ASST Fatebenefratelli Sacco, Milano, Piazza Principessa Clotilde 3, 20121 Milan, Italy; 2Postgraduation School in Radiodiagnostics, Università degli Studi di Milano, Via Festa del Perdono, 7, 20122 Milan, Italy; 3Radiology Department, IRCCS San Raffaele Hospital, via Olgettina 60, 20132 Milan, Italy; 4Unit of Diagnostic Imaging and Stereotactic Radiosurgery, Centro Diagnostico Italiano, Via Saint Bon 20, 20147 Milan, Italy; 5Radiology Department, Fondazione IRCCS Cà Granda, Policlinico di Milano Ospedale Maggiore, Via Sforza 35, 20122 Milan, Italy

**Keywords:** artificial intelligence, lung cancer, deep learning

## Abstract

Lung cancer is one of the malignancies with higher morbidity and mortality. Imaging plays an essential role in each phase of lung cancer management, from detection to assessment of response to treatment. The development of imaging-based artificial intelligence (AI) models has the potential to play a key role in early detection and customized treatment planning. Computer-aided detection of lung nodules in screening programs has revolutionized the early detection of the disease. Moreover, the possibility to use AI approaches to identify patients at risk of developing lung cancer during their life can help a more targeted screening program. The combination of imaging features and clinical and laboratory data through AI models is giving promising results in the prediction of patients’ outcomes, response to specific therapies, and risk for toxic reaction development. In this review, we provide an overview of the main imaging AI-based tools in lung cancer imaging, including automated lesion detection, characterization, segmentation, prediction of outcome, and treatment response to provide radiologists and clinicians with the foundation for these applications in a clinical scenario.

## 1. Introduction

With more than 2 million cases per year in the USA, lung cancer is one of the malignant tumors with the fastest rate of morbidity and mortality growth, and the leading cause of cancer-related death [1,2]. Since most lung cancers are detected in the middle and late stages of the disease, when few treatment options are still available, the 5-year survival rate for individuals is only 10–20% in most countries [3].

Lung cancer is clinically classified into two main histological groups: small-cell lung carcinoma (SCLC) and non-SCLC (NSCLC). NSCLC, which accounts for roughly 85–90% of cases and is the most common type of lung cancer, includes several subtypes: adenocarcinoma, squamous cell carcinoma, large cell carcinoma, and squamous adenocarcinoma [4]. The International Association for Lung Cancer Research classifies lung cancer into I–IV stages based on tumor diameter, lymph node metastasis, and distant metastasis, with stage I–II being the early stage and stage III–IV being advanced lung cancer [5]. The majority of lung cancers are diagnosed at an advanced stage and have a poor prognosis. Furthermore, limitations in treatment selection and prognosis evaluation have presented challenges to clinicians.

To reduce the mortality rate of lung cancer, early diagnosis and appropriate treatment are key factors. So far, lung cancer has mainly been diagnosed with computed tomography (CT) and tissue biopsy, but the information provided on the patient’s prognosis and response to therapy is limited. For these reasons, early diagnosis through individualized screening programs and non-invasive characterization of the lesions thanks to new imaging biomarkers are essential for planning the most appropriate management [6].

In this scenario, the application of artificial intelligence (AI) algorithms has gained popularity in various tasks related to lung cancer imaging such as the improvement of screening programs, lesion detection, characterization, and the prediction of response to therapy and prognosis. Together, these efforts converge toward offering patients truly tailor-made management [7].

### A Quick Introduction to Artificial Intelligence, Machine Learning, and Radiomics

Generally defined as the technology that mimics human intelligence and cognitive processes, such as learning, reasoning, problem-solving, decision-making, and creativity, AI already has a revolutionary but often hidden impact in everyday life. In this section, we will go over the definitions and theoretical frameworks of the most important AI-related concepts in biomedical imaging.

Along with the awareness that radiological images should be considered numerical data rather than mere pictures, AI-based applications in radiology have grown in popularity over the last few decades [8,9]. Within this scenario, the so-called radiomic paradigm is based on the extraction of quantitative and ideally reproducible information from diagnostic images, including complex patterns that are difficult to recognize or quantify by the human eye [10]. According to this definition, radiomics can be considered synonymous with quantitative imaging [11]. 

The radiomic approach exploits sophisticated AI-based algorithms to extract and analyze large quantitative metrics from medical images that, alone or in combination with demographic, histological, genomic, or proteomic data, can be used for clinical problem-solving [10]. Radiogenomics could be considered a subset of radiomic applications, aiming to link imaging and biology, correlating lesion imaging phenotype (“radio“) to the genotype (“genomics”), based on the assumption that phenotype is the expression of genotype [12].

In the very general context of AI, machine learning (ML) represents the backbone of the radiomics approach. ML is a learning paradigm, a set of models and algorithms structured within a precise theoretical framework, aiming for the automated detection of meaningful patterns in data [13,14,15,16]. There are four main types of ML, depending on the level of data pretreatment and the nature of the “feedback” available for the system to infer the relationship between data: supervised learning (SL), unsupervised learning (UL), semi-supervised learning (SSL), and reinforcement learning (RL) (Table 1). Each approach can be used to address different clinical tasks. We will focus mainly on the first two, which are the most used in radiomics research [13].

To explain how the SL works, let’s start by considering a practical problem. Much of the radiologist’s activity consists of very general classification tasks (for example, in lung cancer screening, deciding whether a nodule is benign or malignant). SL is a common ML approach particularly suited to tackling “simple” classification tasks in which data could easily be categorically tagged [13,14,15,16]. To do so, SL algorithms draw inspiration from human learning in a very simple way: learning from examples. 

Usually, a computer is programmed to perform a function (f) on input (x) to obtain output (y). To do this, it is necessary to know exactly (1) the type of operation that the computer has to perform and (2) how to translate this information into a language that the machine can understand (coding) [13,14,15,16]. Hence the problem, from a computational point of view, of the impossibility of translating step by step into the code the extremely complex cognitive process that underlies the diagnostic process of a radiologist. This challenge can be addressed through an ML approach, that is, by leveraging a form of AI. In SL, the learning phase of the model would require a dataset of tuples (x, y) that includes typical examples of inputs (e.g., chest X-rays) and corresponding outputs already labeled by an expert (e.g., chest X-rays already classified by an experienced radiologist as “positive” or “negative”) [13,14,15,16]. By feeding the model with enough examples, the algorithm can infer the relationship that maps the data with a required level of accuracy. After being trained and validated with an external dataset, the model can be used to assist the radiologist in a new diagnostic task: providing the raw unlabeled data (x), it will return a probability estimate that (y).

Considering images as a set of numerical data means making them available for formal processing that would be otherwise impossible with a human-based qualitative approach. UL relies solely on the intrinsic structure of data that has not been labeled, classified, or categorized by an expert [16]. In this kind of ML, the naive dataset is given to the learning algorithm which is then asked to extract knowledge from it, discovering hidden patterns in data, as in clustering tasks, where the aim is to divide the dataset into groups based on specific feature characteristics, or association task, where the aim is to find association rules within the dataset. In contrast to supervised learning, UL exhibits self-organization. 

SSL approach works mostly on unlabeled data, with a small amount of labeled data, thus this type of ML falls between UL (with no labeled training data) and SL (with only labeled training data). This type of ML addresses the problem of low data availability by taking advantage of the abundant amount of accessible but unlabeled (undiagnosed) data in order to train precise classifiers [17]. Finally, RL is a more complex and challenging method and it is not currently used in medical imaging. It learns how to solve a task via interaction and feedback, or in other words by trial and error. Basically, the algorithm is programmed with a goal and a set of rules, and it tends to maximize the rewards or reinforcement it receives from the environment. 

Each ML approach comprises different algorithmic strategies, or models, to map the relations within data, which can be probabilistic methods or neural network (NN)-based methods. Each model represents a different set of rules for manipulating inputs according to a different approach to the problem and a different theoretical background. Popular examples of models used in SL are decision trees, random forest, logistic regression, and support vector machines. Some models can be applied with some modifications, in both SL and UL tasks. 

Among different models available, artificial neural networks (ANNs) can be used with some modifications in both SL and UL, depending on the task. ANNs have proved to be particularly suitable for image analysis, including computer-aided detection (CAD) tools, image segmentation, image generation, etc. [18]. ANNs consist of a biologically inspired programming paradigm in which information is analyzed by several layers of interconnected “nodes” or “cells”. 

Deep learning (DL) is a subdomain of ML that exploits particularly complex ANNs architectures such as convolutional neural networks (CNN), the method of choice for processing visual data. These networks are made of many hidden intermediate layers representing an increasing level of abstraction. CNN could discover intricate patterns in large data sets going beyond the features extracted by radiologists [19]. DL algorithms are often at the base of the radiomic and radiogenomic approaches in cancer imaging. When compared to most ML algorithms, the DL model performance increases dramatically when analyzing large amounts of data, making them particularly suitable for exploratory data analysis and image processing [20]. 

Contrastive learning is a programming paradigm used to train deep classification models for image recognition activities that have seen a resurgence in recent years [21]. There are basically two forms of contrastive learning: supervised contrastive learning and self-supervised contrastive learning. The common idea of these models is to contrast a reference image called an “anchor” and one or more “positive” examples, to a set of “negative” samples. The self-supervised contrastive learning contrasts a single positive example for each anchor (i.e., an augmented version of the same image) against a set of negatives. The supervised contrastive method, on the other hand, contrasts the set of all samples from the same class as positives, against the negatives from the remainder of the batch. 

In this narrative review, we provide an overview of AI contributions to lung cancer radiology, concerning both imaging tasks as automated lesions detection, characterization, and segmentation, and clinical tasks as an imaging-based prediction of outcome and treatment response, to provide radiologists and clinicians the foundation for their applications in a clinical scenario.

## 2. Lung Cancer Screening and Detection

Due to the subtle or no symptoms caused by early-stage lung cancers, the risk of diagnostic delay is high and leads to a poor prognosis, with a 5.2% 5-year survival rate for advanced stages [22]. Early diagnosis, on the other hand, is proven to dramatically reduce lung cancer-related mortality, with a 5-year survival rate of 57.4% [23]. Several randomized controlled trials based on chest X-rays, with or without sputum cytology, have been conducted to screen the population at high risk of developing lung cancer, showing that screening led to early diagnosis but not a reduction in cancer-related mortality [24,25,26,27,28,29]. In contrast, low-dose CT (LDCT) screening has established itself as an effective screening test, capable of reducing lung cancer mortality [30,31,32].

Based on this evidence, the United States Centers for Medicare & Medicaid Services (CMS) established that patients aged 55–77 with a 30-pack-year smoking history are eligible for CT screening programs, although new guidelines suggest that the target populations should be expanded further [33,34]. In 2006, the European Union approved a position statement in support of a risk-based implementation of LDCT lung cancer screening [35]; however, population-scale screening programs are lacking and, in clinical practice, only a small proportion of eligible patients are screened, due to the excessive workload and difficulties in documenting actual tobacco exposure [36,37,38]. AI techniques can allow for processing and integrating large volumes of data and extracting meaningful information to direct the screening decision process [39].

Using data from electronic patient records (chest X-ray image, age, sex, current smoke), Lu et al. created a CNN model for the prediction of the long-term incidence of lung cancer. The model demonstrated superior discrimination for incident lung cancer than CMS eligibility criteria when compared to screening groups of the same size (area under the curve (AUC), 74.9% vs. 63.8%, *p* = 0.012), and 30.7% fewer tumors were missed [40]. 

In another study, Gould et al. developed an ML model to predict a diagnosis of lung cancer based on clinical and laboratory data. Its accuracy was compared to the modified version of the Prostate, Lung, Colorectal, and Ovarian Cancer Screening Trial risk model (mPLCOm2012). The model showed higher accuracy than the mPLCOm2012 in detecting NSCLC 9–12 months before clinical diagnosis (*p* <0.001) and higher accuracy than standard eligibility criteria for lung cancer screening and mPLCOm2012 when applied to a screening-eligible population [41]. 

Although promising, the use of AI in risk stratification is still in its early stages, and more studies are needed to determine its true clinical impact. 

The development of CAD systems represents a valuable contribution of AI to lung cancer screening, and several tools have been validated and are currently available on the market (Figure 1) [42,43]. Automatic nodule detection is performed to identify structures within the lung that may be malignant nodules. DL models emerged as particularly well-suited for screening applications, but they must be trained and tested on high-quality datasets in order to realize their full potential. Some open-source image repositories for lung cancer are available and partially meet this need [44]. The most used databases are The Lung Image Database Consortium and Image Database Resource Initiative (LIDC-IDRI), which comprises 1018 CT scans and 36,378 lung nodules [45], the extensive Lung Nodule Analysis 16 (LUNA 16) dataset, derived from LIDC-IDRI, which includes 888 selected CT scans and 13,799 lung nodules [46], and the Ali Tianchi dataset, which includes information on 1000 CTs and 1000 nodules. 

Many studies explored the performances of DL-based tools for pulmonary nodule detection (Table 2). Chi et al. [47] developed a new deep CNN framework made up of three cascaded networks for detecting pulmonary nodules in chest CT scans. Its performance was tested on the LUNA16 and Ali Tianchi datasets [45] as the training and testing samples, with precision, sensitivity, and specificity of 0.8792, 0.8878, and 0.9590, respectively. 

Nasrullah et al. [48] used two deep three-dimensional customized mixed link network (CMixNet) models for lung nodule detection and classification. The system was evaluated on the LIDC-IDRI dataset with high sensitivity (94%) and specificity (91%).

Kopelowitz et al. [49] implemented the MaskRCNN model, a system for 2D object detection and segmentation, to handle three-dimensional images for nodule detection and segmentation on CT scans, with a sensitivity of 93%.

Ding et al. [50] developed a modified Faster R-CNN for the detection of malignant pulmonary nodules. The performance on the LUNA 16 dataset demonstrated a sensitivity of 94%. 

Khosravan et al. [51] applied a 3D CNN called S4ND based on the Single-Shot Single-Scale Lung Nodule Detection system to detect lung nodules without further processing, with a sensitivity of 95.2 %. Cai et al. [52] used Mask R-CNN architecture as a backbone and applied a feature pyramid network to extract feature maps. A region proposal network was then used to generate bounding boxes for candidate nodules from the generated feature maps. The results were validated through the LUNA16 dataset, achieving a sensitivity of 88.70%. 

When deciding whether to include models in standard clinical practice, the potential economic impact of AI systems is a key consideration. A study by Ziegelmayr et al. investigated a previously proposed 3D-CNN developed by Ardila et al. [53] for the analysis of LDCT in a baseline lung cancer screening and demonstrated the possibility of costs reduction and increased effectiveness through the use of AI [54].

**Table 2 diagnostics-12-02644-t002:** The table lists the characteristics of different studies aiming at lung nodules screening and classification.

Authors	Country	Imaging Modality	Patient Number	Study Model	AI System	Validation	Main Theme	Strengths	Weakness
Nasrullah et al. [48]	China	LDCT	LIDC-IDRI dataset	Retrospective	Two deep 3D customized mixed link network architectures for lung nodule detection and classification	LIDC-IDRI and LUNA 16 dataset	Lung nodule detection and classification	The system achieved promising results in the form of sensitivity (94%) and specificity (91%)	Validation only in pre-clinical settings
Kopelowitz et al. [49]	U.K.	CT	LUNA 16 dataset	Retrospective	Modified MaskRCNN to handle 3D images	LUNA 16 dataset	Lung nodule detection and segmentation	All-in-one system for detection and segmentation	Validation only on the LUNA 16 dataset
Ding et al. [50]	China	CT	LUNA16 dataset	retrospective	Faster R-CNN for detection and three-dimensional DCNN for the subsequent false positive reduction	LUNA16 dataset	Lung nodule detection	Good detection performance on nodule detection ranking the 1st place of Nodule Detection Track (NDET) in 2017	Needs validation on bigger datasets
Khosravan et al. [51]	U.S.A.	CT	LUNA16 dataset	retrospective	3D densely connected CNN	LUNA16 dataset	Lung nodule detection	single-shot single-scale fast lung nodule detection algorithm without the need for additional FP removal	Validation only on the LUNA 16 dataset
Tran et al. [55]	Vietnam, France	CT	LUNA16 dataset	retrospective	15-layer 2D deep CNN architecture (LdcNet)	LUNA16 dataset	automatic feature extraction and classification of pulmonary candidates as nodule or non-nodule	High-quality classifier with an accuracy of 97.2%, sensitivity of 96.0%, and specificity of 97.3%.	Only validated in one preclinical dataset
Wu et al. [56]	China, U.S.A., Australia, U.K., Germany	CT	LIDC-IDRI dataset	Retrospective	50-layer deep residual network	LIDC-IDRI dataset	Lung nodule classification	The lung nodule image can be used as the input data of the network directly, avoiding complicated feature extraction and selection.	Long training time is needed when dealing with a large number of lung CT images
Mastouri et al. [57]	Tunisia	CT	LUNA16 dataset	Retrospective	Three bilinear-CNN followed by a linear SVM classifier	LUNA16 dataset	Lung nodule classification	The system was validated on the LUNA16 dataset and compared to the outcomes of conventional CNN-based architectures showing promising and satisfying results	The bilinear pooling requires massive calculation and storage costs, making this algorithm impractical
Al-Shabi et al. [58]	Malaysia, Singapore, U.S.A.	CT	LIDC-IDRI dataset	Retrospective	Gated Dilated(GD) network	LIDC-IDRI dataset	Classification of pulmonary nodules as benign or malignant	Better discrimination whether benign or malignant for mid-sized nodules	Requires an object detector model to identify the nodule locations before classifying them as benign/malignant
Liu et al. [59]	China, U.S.A.	CT	LIDC-IDRI dataset	Retrospective	multi-model ensemble learning architecture based on 3D convolutional neural network (MMEL-3DCNN)	LIDC-IDRI dataset	Benign/malignant lung nodule classification	Image enhancement on the input data to improve the contrast of lung nodules with low contrast to surrounding tissues	Validation only in pre-clinical setting

## 3. Lung Nodule Classification

One of the main limitations of CAD systems is the high false positive rate related to the presence of blood vessels and other soft tissue structures, which results in reduced accuracy and efficacy of CAD screening tools in large populations [60].

Adopting effective classification techniques can reduce false positives and significantly improve the accuracy of nodule identification.

Tran et al. [55] used a novel 15-layer 2D deep CNN model for automatic feature extraction and classification of pulmonary candidates as nodules or non-nodules, with an accuracy of 97.2%, sensitivity of 96.0%, and specificity of 97.3%. 

Wu et al. [56] developed a deep residual network to classify lung nodules that was built by combining residual learning and migration learning. The proposed approach was verified on lung CT images from the LIDC-IDRI database, reaching an average accuracy of 98.23% and a false positive rate of 1.65%.

Mastouri et al. [57] proposed a bilinear CNN (BCNN) consisting of two-stream CNNs (VGG16 and VGG19) as feature extractors combined with a support vector machine classifier for false positive reduction. They found that the BCNN (VGG16 + VGG19) combination with and without support vector machine surpassed the single VGG16 and 19 models, achieving an accuracy rate of 91.99% and an AUC of 95.9%.

Once the nodularity has been identified as a proper nodule, the next step is represented by the differentiation between benign and malignant pulmonary nodules, improving overall diagnostic efficiency. For this purpose, CNNs are applied to extract and analyze different features of lung nodules, such as shape, growth rate, and morphology [44].

Zhang et al. [61] explored the use of the DenseNet architecture with 3D filters and pooling kernels. The performance of the proposed nodule classification was evaluated on the LUNA16 dataset achieving 92.4% classification accuracy. 

Al-Shabi et al. [58] proposed a novel CNN architecture called Gated-Dilated to classify nodules as malignant or benign. The system was evaluated with the LIDC-LDRI showing accuracy of 92.57% and an AUC of 0.95.

Liu et al. [59] developed a multi-model ensemble learning architecture based on a 3D CNN, which was tested on LIDC-IDRI with a sensitivity of 90% and false positivity of 30%. The features of the reported studies are outlined in Table 2. 

## 4. Segmentation

Lung CT image segmentation (Figure 2) is a critical process in many applications, including lung cancer diagnosis, accurate disease burden definition, correct and reproducible extraction of radiomic features, and objective evaluation of treatment response. The automated identification and segmentation of lung nodules have a significant impact on lung cancer treatment and patient survival [62]. The most important segmentation task is the figure/background resolution [63,64]. Despite not being provided to radiologists in real scenarios, an accurate lung mask is essential for the development of clinical support tools to avoid the inclusion of noise and non-relevant background information [65]. Segmenting the lung fields is challenging due to the inhomogeneity of the lung volumes [66]. The most popular lung segmentation approach uses so-called hand-crafted characteristics to successfully differentiate the regions of interest from each other. Segmentation techniques can be classified into the following types: threshold, edge-detection, region growing, deformable boundary, and deep learning models. The threshold-based method is based on the principle that normal lung tissue has a lower density than other tissues; thus, the lung can be distinguished by applying a mask with a specific density threshold [67]. Although this is the most commonly used method, it has several limitations, including the inability to remove the trachea and main bronchi, and the heterogeneity of acquisition protocols, which makes it impossible to set a universal gray-level threshold for segmentation [68,69]. 

The edge detection technique uses image processing to define boundaries between different regions based on distinct gray surface properties. The main drawbacks of this method are its sensitivity to noise and its inability to work on images with smooth transitions and low contrast [63]. 

The region-based segmentation groups pixels/voxels with homogeneous properties according to a predefined criterion. The most common segmentation method belonging to this technique is the region-growing method which begins with the positioning of a “seed” by the reader that gradually grows and automatically adds neighboring pixels that respect a similarity criterion such as color, intensity, or texture [70]. The advantages are low computational complexity and high speed, whereas the main disadvantages are sensitivity to noise or variation in intensity. Moreover, this method does not allow to segment nodules attached to the pleura [71]. 

The deformable boundary method takes into account the entirety of object margins and can include prior information about the object’s shape. An active contour model or snake is the most commonly used tool. These models frequently require human interaction during the initial contour construction [72]. 

The last segmentation method assigns each pixel/voxel to a specific class for different areas of the image. In this method a natural dataset can be processed in its raw form by segmentation techniques based on DL algorithms, overcoming the limitations of hand-crafted features. A relevant example of fully automatic segmentation is the system proposed by Long et al. based on a fully convolutional network in which transposed convolutional layers replace the last fully connected layers of CNNs. Ronneberg et al. developed the U-shape Net (U-Net) for biomedical segmentation tasks based on this concept [73]. U-Net is made up of two processes: a contracting path for capturing context and a symmetric expanding path for accurate localization. The U-Net model has several important advantages; in particular, it works with very little data, it can use global location and context information at the same time, and it ensures that the input images have full texture. The main disadvantage of the U-Net model is the two-stage process, which involves applying separate processing steps to each group feature map before concatenating the feature maps. Various extensions of U-Net have been developed in recent years [74], with the ResNet34 pre-training model being used in its contraction path. The most significant benefit of this modification is increased training speed and network extension power. In another study, U-Net was extended to a network called BCDU-Net [75], which outperformed modern alternatives for medical image segmentation. Bhattacharyya et al. recently proposed a new method based on U-Net for lung nodule segmentation, a weighted bidirectional feature network employed to create a modified U-Net architecture, called DB-NET: in this way, they obtained better performance in the segmentation of cavitations, ground glass opacities, tiny, and juxta-pleural nodules with a Dice coefficient of 88.89 ± 11.71, superior to compared methods [76].

## 5. Prediction 

In recent years, oncology has seen an ever-increasing variety of therapeutic options and an attempt to gradually transition toward personalized medicine. In particular, lung cancer, owing to its high prevalence and extensive research, can boast a number of active and up-and-coming targeted treatment options. As a result, it becomes increasingly important to match the right patient to the right treatment through detailed characterization of both the tumor and the patient. 

The identification of prognostic biomarkers that provide information about the likelihood of a disease-related endpoint can allow establishing of the patient’s risk profile based on tumor characteristics and identifying patients with a poor prognosis who may be candidates for therapy escalation and/or enrollment in experimental trials [77,78].

AI-enhanced radiology assessment of qualitative and quantitative features provides a continuously increasing contribution to this in-depth assessment. 

Characterization of gene expression patterns of lung tumors is one of the mainstays of profiling for targeted therapy choice. The most frequently mutated oncogenes and primary molecular targets in NSCLC include epidermal growth factor receptor (EGFR), Kirsten rat sarcoma viral oncogene (KRAS), and anaplastic lymphoma kinase (ALK) [79]. Tyrosine kinase inhibitors (TKIs), such as gefitinib and erlotinib, are effective in treating EGFR-mutated tumors, but not KRAS-mutated tumors, despite being part of the same signaling pathway. ALK rearrangement tumors, however, are sensitive only to TKIs that specifically target ALK, such as crizotinib.

Targeted TKIs improve survival and reduce drug-induced toxicities in patients with NSCLC compared to standard chemotherapeutic agents [80]. Some of these genetic alterations can be inferred through standard imaging characteristics. For example, the proportion of ground–glass opacity is predictive of EGFR mutations in lung adenocarcinomas [81,82].

With the use of AI algorithms, EGFR status could be predicted by combining clinical and CT features such as smoke history, tumor size, bubble-like lucency, enhancement pattern, presence of pleural retraction, and thickened adjacent bronchovascular bundles (AUC = 0.778) [83].

Radiomics and radiogenomics allow for further bridging of the gap between imaging data and the biological characteristics they represent [84,85]. Radiomics allows non-invasive assessment of tumor behavior and phenotype that goes beyond what can be directly identified by human operators. Radiogenomics focuses on the extraction of quantitative imaging features that correlate to tumor expression patterns and identify target genes and pathways. As opposed to biopsy, radiogenomic assessment is not limited by tumors’ location or heterogeneity of bioptic material and can assess multiple neoplastic lesions at different time points. As such, radiogenomics represents an attractive non-invasive, repeatable, and cost-effective alternative to traditional methods of genetic and molecular profiling of tumors [80,86]. A radiomics feature extracted from the pretreatment CT scan of NSCLC patients showed a strong association with the presence of sensitizing EGFR mutations (AUC = 0.67, *p* = 0.03); in contrast, tumor volume and maximum diameter were both not significantly predictive of EGFR mutations (*p* > 0.27) [80]. Another radiomics model reliably predicted EGFR mutations in NSCLC based on the presence of emphysema, airway abnormalities, the percentage of the ground glass component, and the type of tumor margin (AUC = 0.89) [87]. A combined radiomic and clinical model reached an AUC of 0.76 to predict the most common subtype of EGFR mutation (L858R) [88]. A radiogenomic CT-based biomarker consisting of tumor location, pleural effusion, and pleural tail sign showed a strong correlation with ALK rearrangements with a sensitivity of 83.3% and specificity of 77.9% [89]. A DL model based on CT data combined with clinicopathological information assessed the ALK fusion status in NSCLC patients with an AUC of 0.8481 and predicted response to ALK-specific TKI therapy [90]. Another ML model with integrated radiomics features predicted ALK mutation status with an AUC of 0.83. The addition of conventional CT and clinical information to the model did not result in significant performance improvement [91]. A complex prediction model using clinical data and radiomics features from CT and PET studies identified ALK/ROS1/RET fusion-positive status associated with an important response to ALK inhibitors in pulmonary adenocarcinomas [92].

Radiomics signatures can also allow the optimization of other therapies. In SCLC patients, a complex radiomics signature was significantly associated with the efficacy of first-line chemotherapy consisting of etoposide and cisplatin (*p* < 0.05). The performance of the radiomics signature to predict the chemotherapy efficacy (AUC = 0.797) was better compared with the model using clinicopathological parameters (AUC = 0.670) [93]. Similarly, radiomics signatures can predict the response to neoadjuvant chemotherapy based on pre-treatment CT in NSCLC patients [94,95]. In lung cancer patients who underwent radiotherapy, radiomic features obtained from 3D maps allowed the prediction of acute and late pulmonary toxicities [96].

Immunotherapy is a novel approach to cancer treatment that exploits the tumoral environment through targeted activation of immunological synapses. Although immunotherapy has shown the potential to improve the prognosis of lung cancer patients with remarkable survival outcomes, its use is still limited by high costs and toxicities. Moreover, its clinical benefit has so far been limited to only a subset of patients, most notably to those with PDL-1-positive expression. It is thus important to guarantee comprehensive pretreatment assessment and correctly identify early responders in order to optimize the cost-effectiveness and clinical impact of immunotherapy [97]. The expression of PDL-1 on immunohistochemistry is a biomarker routinely used to select immunotherapy candidates [98]. However, bioptic samples can be limited by the spatial and temporal heterogeneity of the tumors, as well as the feasibility and invasiveness of the biopsy. AI markers based on integrated imaging data provide a complementary solution for baseline evaluation of immunotherapy candidates and assessment of response on follow-up imaging. The expression of PDL-1 level can be assessed using various radiomics features [99,100]. Changes in a CT radiomic indicator associated with the density of tumor-infiltrating lymphocytes throughout the immunotherapy could identify early responders with an AUC of 0.88 ± 0.08 [101]. Another model could predict a response to anti-PD-1 immunotherapy based on pre-treatment CT in NSCLC with an AUC up to 0.83 [102]. Prediction of immunotherapy-related toxicity represents another area of the potential application of AI algorithms in lung cancer. For example, Mu et al. developed a radiomics model that could predict immune-related adverse events among patients with advanced NSCLC treated with immunotherapy. The model included a radiomics score based on FDG-PET/CT images, a type of immune checkpoint inhibitor, and a dosing schedule [103].

AI prediction models can successfully estimate the prognosis of lung cancer patients through the identification of complex imaging biomarkers and their integration with clinical data. Pre-treatment radiomic signatures were significantly associated with survival in patients with NSCLC. In particular, features describing tumoral heterogeneity were associated with worse survival in all datasets [104]. In another study, quantitative features associated with lower convexity (i.e., presence of speculated margins) and higher entropy gradients (i.e., intratumor density variation) in lung adenocarcinomas were strongly associated with worse prognosis in patients with early-stage lung cancer [105]. A prediction model combining pretreatment radiomics tumor parameters with immune parameters such as PDL-1 expression and density of tumor-infiltrating lymphocytes and CD3 expression identified a favorable outcome group characterized by a favorable immune-activated state [106]. A CT-based DL system predicted progression-free survival and identified features associated with the TKI-resistant EGFR genotype [107]. A model with DL radiomics features and integrated circulating tumor cell count could predict the recurrence of early-stage NSCLC patients treated with stereotactic body radiation therapy [108]. Moreover, in NSCLC patients with brain metastases, radiomics features allow distinguishing subgroups with different survival durations [109]. Although less information is available on SCLC, one study identified a radiomics model that combined 11 features from lung and mediastinal CT windows and allowed for predicting the progression-free survival of small cell lung cancer patients with an AUC of 0.8487. Another algorithm based on CT radiomics features did not show a correlation with overall and progression-free survival of SCLC patients, nor did it correlate with the expression of immunohistochemical markers [110]. Lian et al., in a population study of 1705 patients with lung cancer in stages I and II, used clinical (e.g., age, histologic type, tumor location) and imaging data to develop an effective model, able to predict overall and recurrence-free survival in early lung cancer stages [111].

The main characteristics of the above studies are listed in Table 3.

## 6. Challenges

The advent of AI in biomedical imaging may have a potentially revolutionary impact on a variety of activities, ranging from early diagnosis to prognosis and lung cancer treatment planning. Once implemented in clinical practice, this will lead to a significant improvement in patient management. However, the widespread adoption of AI-based tools in the daily work routine is currently hampered by several obstacles.

The development of AI-based tools needs a large amount of high-quality data. Although lung cancer datasets are widely available, imaging, clinical, and laboratory data should be collected in a highly standardized and well-organized manner to allow the development of robust algorithms [113]. The collection of data from multiple institutions is therefore desirable, hence the importance of collaboration and data sharing in research. Open-access image repositories such as “The Cancer Imaging Archive”, which includes a large dataset on cancer, are still expanding and represent a useful aid for researchers in order to validate local studies [114].

Collaboration is a key theme in AI research for medical purposes, as the development of effective models with a real impact on daily clinical practice requires the effort of multidisciplinary teams, including radiologists, physicians, engineers, and software developers, who share their knowledge in a mutually beneficial multidirectional manner.

Another important limitation is related to the study design. In particular, a number of studies investigating outcome prediction in lung cancer only analyzed small cohorts of patients [115,116,117,118,119,120]. The results of AI models trained with small case series are difficult to generalize and therefore not applicable in real-life clinical practice. Further validation in external cohorts is required to determine the reliability and clinical usefulness of the results. Similar limitations are applied to models developed through retrospective data, which must be tested in a prospective scenario before they can be used as clinical diagnostic aids.

Furthermore, when designing an AI model, researchers should be aware that multiple data sources would have to be incorporated to fully characterize lung cancer. To create a more comprehensive model, the guidelines recommend performing multivariable analysis with non-imaging features, including family history and clinical and genetic data to develop holistic models [121].

Reproducibility is one of the main challenges that AI must overcome to achieve clinical implementation, as there can be numerous differences in every aspect of the radiomics workflow between different studies and research institutions, from image acquisition to model validation [122]. For example, the heterogeneity of image acquisition protocols among different institutions can affect the signal-to-noise ratio and the characteristics of extracted images. This implies that variations in imaging features and values between patients may be due to acquisition parameters rather than variations in tissue biology [122]. The exclusion of features strongly influenced by acquisition and reconstruction parameters can represent a strategy to overcome this limitation [123]. Another solution could be to improve the standardization of image acquisition, for example by using open imaging protocols. 

According to Lambin et al., [121] the widespread availability of medical imaging data has created an environment ideal for the quick development of ML and data-based science. Radiomics-based models may represent a powerful tool for precision medicine, but standardized data collection, evaluation criteria, and reporting guidelines, as well as valuation of both the scientific integrity and the clinical relevance of research study, are required for further development of radiomics–based systems. The authors, therefore, proposed a quality score based on 16 components of the radiomics workflow to define the robustness of radiomics studies. 

Stability measures can be used to build more reproducible radiomic models: Khorrami et al. combined stability and discriminability criteria in developing radiomic classifiers to predict disease recurrence in early-stage NSCLC cancer on CT images from 610 patients of four independent cohorts [112].

The segmentation procedure can represent a confounder factor in AI protocols: radiomics usually needs precise delineation of tumor boundaries and outlines to allow the computation of lesion characteristics such as size, shape, heterogeneity, and the accurate extraction of feature data from the segmented volumes [124]. Manual segmentation has been proposed as the ground truth, but it is affected by high interobserver variability [125] and is a time-consuming procedure, unfeasible for large image data sets. The development of accurate semiautomatic segmentation procedures with minimal user interaction, reproducibility, and time efficiency can represent a solution to this issue. Additionally, automatic segmentation has been tested with promising results, but validation on larger series must be conducted to obtain clinical integration [126]. Another crucial point is represented by the algorithms’ transparency and interpretability, moving beyond the black box to ensure the explicability and trustworthiness of the results. Many of these so-called “black-box” approaches may be viable in the diagnostic setting (e.g., AI tools for triaging time-sensitive scans); the issue of interpretability, however, becomes more important when it comes to AI-enabled imaging biomarkers for treatment optimization because a biomarker-driven treatment decision needs an explanation rooted in pathophysiology [77]. The comparison and validation with the clinical gold standard are also needed to prove the clinical utility of the application in everyday practice, and the benefit deriving from the use of a certain algorithm compared to other approaches should be assessed.

These obstacles must be overcome to move toward the clinical implementation of artificial intelligence tools for lung cancer. Guidelines should be developed to guide the development and validation of future AI-based studies, and prospective clinical trials should be carried out to establish the utility of AI tools in clinical practice, their impact on patient care, and overall outcomes. Collaboration, data exchange, and extensive model validation are crucial for AI-wide clinical translation. Legal and ethical concerns about the use of AI in cancer imaging are also potential roadblocks to investigation and implementation. Most research necessitates large medical datasets, involving imaging and clinical data. This raises legal and ethical concerns about who “owns” the data and has the right to use it, especially if commercial value is involved [127]. Explicit patient consent for data use would be ideal, but due to the number of patients included in huge datasets, particularly in retrospective studies, this may not be feasible.

Even if researchers are granted permission to use anonymized or de-identified data, difficulties could arise if identifying information is included in the image or necessary for the investigation [128]. Clearer legal and ethical frameworks, as well as input from all stakeholders, are needed for the widespread use of AI in oncological imaging [129].

## 7. Future Perspectives

Every day more and more imaging and clinical data are available; however, these data are not usually curated in terms of labeling, segmentations, quality assurance, or pertinence with a predetermined pathology. Data curation is one of the major challenges in developing an AI algorithm, as it requires experienced physicians and is time-consuming. This limitation is particularly consistent in methods requiring big data, such as CNN. A solution could be represented by unsupervised and self-supervised approaches that do not need labeling or supervision [130].

Developments in unsupervised learning, such as variational autoencoders and generative adversarial network learning without any explicit labeling are showing promising results [131]. For example, an unsupervised DL approach for chest CT automatic segmentation showed an accuracy of up to 98% [132], whereas a self-supervised approach developed for the segmentation and classification of lung disorders on chest X-rays and tested on the NIH chest X-ray dataset gave interesting results [133]. We believe that AI clinical research will move in these directions. 

The development of platforms allowing interoperability among the numerous AI applications to create a network of powerful tools and integration of AI technology into picture archiving and communication systems is also needed to bring AI into clinical everyday practice.

Artificial intelligence will make it possible to progressively move towards automating repetitive and time-consuming tasks, such as screening for lung nodules and identifying image-based biomarkers. These developments will optimistically allow disease characterization in a non-invasive and repeatable way, improving therapeutic management in order to achieve the goal of personalized medicine (Figure 3) [134].

## 8. Conclusions

AI has the potential to be a real game-changer in the early detection and clinical management of lung cancer.

In terms of early lung cancer diagnosis, treatment, and follow-up, radiological imaging is at the forefront. AI models comprising imaging data may allow the identification of new predictive and prognostic biomarkers and enhance decision-making and outcomes for lung cancer patients. 

Multimodality integration with the collection and combination of multiple sources of information with the creation of holistic models may allow the accurate characterization of lung cancer.

Most published papers have shown promising results for AI in characterizing cancer phenotypes and biology in a noninvasive and reproducible way. 

To establish these systems in clinical practice, though, there are many obstacles to be overcome.

Collaboration among institutions is more important than ever because it is well demonstrated that AI tool development requires extensive data sets, similar to how radiologists, clinicians, and AI experts must collaborate closely to ensure the transition into the clinical scenario.

Guidelines should be developed to guide the structuring and ensure the reliability of AI-based research.

AI capacities are unlimited and its progressive incorporation in the clinic may lead to personalized medicine. 

In the age of AI, radiologists and clinicians must comprehend and become familiar with the current state, possible clinical uses, and challenges of AI in chest imaging.

## Figures and Tables

**Figure 1 diagnostics-12-02644-f001:**
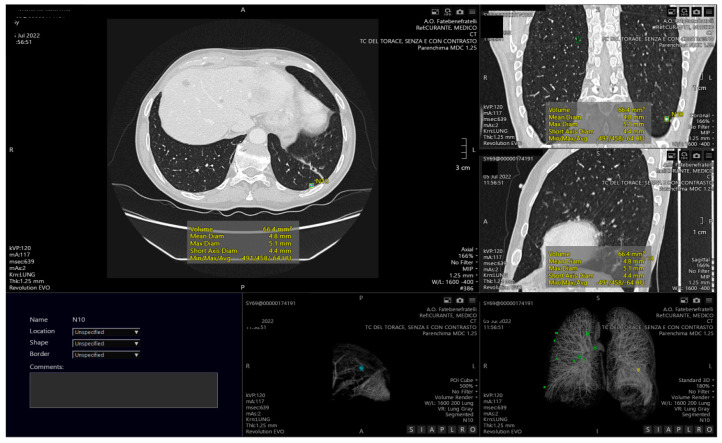
Example of computed aided detection with automatic identification of pulmonary nodules, both visible on axial CT images and in the volumetric reconstruction.

**Figure 2 diagnostics-12-02644-f002:**
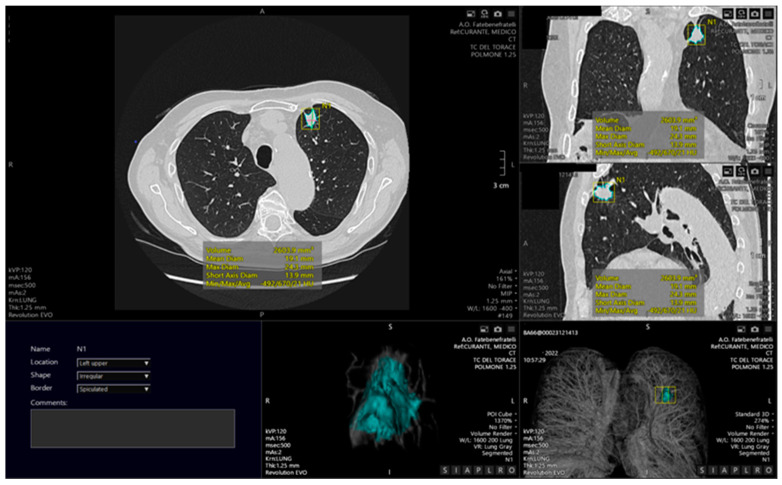
Example of automated segmentation of a pulmonary lesion located in the upper lower lobe, with automated calculation of the lesion volume, visible also as a 3D reconstruction.

**Figure 3 diagnostics-12-02644-f003:**
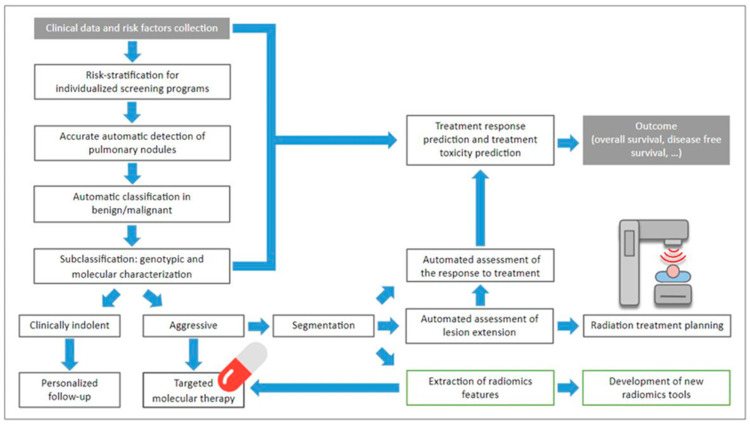
Flowchart showing all the possibilities that AI tools allow for lung cancer, for better and more personalized patient management.

**Table 1 diagnostics-12-02644-t001:** The table describes the different types of ML, the mechanism on which they are based, some examples of possible applications, as well as some operating models for each type [13,14,16,17]. ANNs = artificial neural networks, DBSCAN = density-based spatial clustering of applications with noise, SARSA = state–action–reward–state–action.

Type of ML	Mechanism	Type of Data Provided	Problems that Can Solve	Examples of Models
**Supervised learning (SL)**	The algorithm is provided with tuples of input and output (x,y) and the algorithms infer the relation that maps the dataset	Labeled data	Classification task (discrete variable) Regression task (continuous variable)	Logistic regression Decision Tree Random Forest ANNs
**Unsupervised learning (UL)**	The algorithm exhibits self-organization to capture hidden patterns in data	Unlabeled data	Clustering Association Anomalies detection	Hierarchical clustering K-mean DBSCAN ANNs
**Semi-supervised learning (SSL)**	Falls between unsupervised learning (with no labeled training data) and supervised learning (with only labeled training data; a mix of SL and UL	Mostly unlabeled data, with a small amount of labeled data	Transductive task (infer the correct labels for the given unlabeled data) or inductive tasks (infer the correct mapping from x to y).	Generative model Self-training model Co-training model Transductive model Graph-based model
**Reinforcement learning (RL)**	The algorithm is programmed with a goal and a set of rules. It tends to a nearly optimal policy that maximizes the “reward function” or reinforcement signals	Not needing labeled input/output pairs to be presented, only a numerical performance score is given as guidance.	Economics and game theory under bounded rationality, control theory	Monte Carlo methods Q-learning SARSA methods

**Table 3 diagnostics-12-02644-t003:** Characteristics of different studies on the use of AI in outcome prediction and response to treatment, including strengths and limitations.

Authors	Country	Imaging Modality	Patient Number	Study Nature	AI System	Validation	Main Theme	Strengths	Limitations
Aerts et al. [104]	USA	CT	47	Prospective	Prognostic radiomics signature	Yes	Radiomic data could define a response phenotype for NSCLC patients treated with Gefitinib therapy	Strong associations	- Limited sample size - Only 11 independent radiomic features
Gevaert et al. [87]	USA	CT	186		Predictive radiogenomics decision model		Association between ground glass opacity and the presence of EGFR mutations		Need for validations
Zhao et al. [88]	China	CT	637		Predictive radiomics model	Yes	Radiomics-based nomogram, incorporating clinical characteristics, CT features and radiomic features, can non-invasively and efficiently predict the EGFR mutation status	Sample size	- Different CT scanning parameters - Single center study
Yamamoto et al. [89]	USA	CT	172	Retrospective	Predictive radiogenomics model	Yes	ALK+ tumors have a CT radiophenotype that distinguishes them from tumors with other NSCLC molecular phenotypes	Multi-institutional, international study cohort	- Limited sample - No treatment response validation
Song et al. [90]	China	CT	937	Retrospective	Three blocks deep learning neural network		DLM trained by both CT images and clinicopathological information could effectively predict the ALK fusion status and treatment response		- Small size of the ALK-target therapy cohort (n = 91)
Chang et al. [92]	China	PET/CT	526	Prospective	Three predictive radiomics models		PET/CT-clinical model has a significant advantage to predict the ALK mutation status		- Images acquired and processed in the same way - Single medical center
Wei et al. [93]	China	CT	134	Prospective	Predictive radiomics signature model via binary logistic regression model		The radiomics model (21 features) was superior to clinical model in predicting the efficacy of chemotherapy in patients with SCLC		
Bourbonne et al. [96]	France		167	Retrospective	Three predictive radiomics models via neural network training		In patients with lung cancer treated with RT, radiomic features extracted from 3D dose maps seem to surpass usual models based on clinical factors and DVHs in predicting APT and LPT		
Jiang et al. [99]	China	PET/CT	399		Predictive radiomics models via logistic regression and random forest classifiers	Yes, five-fold cross-validation	Imaging-derived signatures could classify expression rate of specific PD-L1 type		- Stage IV NSCLC patients composed a very small proportion - PET/CT data were obtained in clinical routine through two different manufacture-derived machines with different scanning parameters
Yoon et al. [100]	South Korea	CT	153	Retrospective	Two predictive radiomics model via multivariate logistic regression		Quantitative CT radiomic features can help predict PD-L1 expression		- Patients were identified only from those having PD-L1 testing results - Proposed prediction model did not undergo external validation - PD-L1 test lacks universal reference standards
Khorrami et al. [112]	USA	CT	139	Prospective	Machine learning-based radiomics texture features (DelRADx)	Yes	DelRADx features were (1) predictive of response to ICI therapy, (2) prognostic of improved overall survival, and (3) associated with TIL density on corresponding diagnostic biopsy samples	- Validation in two independent test sets - Radiomic features extracted also from the annular perinodular regions	- The sizes of cohorts, both for discovery and validation, were relatively small - Radiomic feature expressions might be sensitive to lesion annotation accuracy
Trebeschi et al. [102]	Netherlands	CT	203		Machine learning-based radiomics model	Yes	Higher levels of surface-area-to-volume ratio in nonresponding lesions in both cancers suggest that more compact and spherical profiles are associated with better response	Individual lesion-based approach, avoiding the issue of mixed response	Need for validation in larger cohorts
Grove et al. [105]	USA	CT	109	Retrospective	Predictive CT-based features: convexity morphological feature and		Quantitative imaging biomarkers can be used as an additional diagnostic tool in management of lung adenocarcinomas.	Development of imaging features that were descriptive and reproducible using retrospectively acquired clinical scans	- Cohort sample sizes - The two cohorts are likely not comparable (different overall survival trend)
Tang et al. [106]	USA	CT	190	Retrospective	Immunopathology-informed model (IPIM)	Yes	First radiomics model to leverage immunopathology features (CD3+ cell density and percent tumor cell PDL1 expression) to obtain immune-informed radiomics model yielded subtypes associated with OS		- Conducted at a single institution
Wang et al. [107]	USA/China	CT	18232	Prospective	Fully automated artificial intelligence system (FAIS)	Yes	FAIS learned to identify patients with an *EGFR* mutation who are at high risk of having TKI resistance		- Other genes are relevant to targeted therapy (e.g., ALK, KRAS) - Combined method (whole lung + tumor-based) wasn’t studied
Jiao et al. [108]	USA	CT	421		Convolutional AE DL model with three layers of CNNs		Integrating DL radiomics models and CTC counts improves patient stratification in predicting recurrence outcomes for patients treated with SBRT for ES-NSCLC		

## Data Availability

This is a review article, we did not use any research data and we did not provide any result, but just a description of the updated literature on the topic.

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
