# Peer review of "Artificial Intelligence in Lung Cancer Imaging: Unfolding the Future"

_diagnostics, 2022, doi:10.3390/diagnostics12112644_

Round 1
Reviewer 1 Report
The authors provided a review article regarding AI used in detection and diagnosis of lung cancer. However, these are my initial comments that the authors should consider:
1. Create tables identifying the strengths and weaknesses of the reviewed studies.
2. Additional figures and tables showing the features and performance of the specified studies being compared to each other.
3. What will be the possible studies in the future? It can be represented by roadmap specifying the years within 10 years to which the first AI technique 10 years ago was implemented for lung cancer up to the recent work.
Author Response
The authors thank you very much for your work and precious suggestions.
We Performed the required changes as follows:
The authors provided a review article regarding AI used in detection and diagnosis of lung cancer. However, these are my initial comments that the authors should consider:
- Create tables identifying the strengths and weaknesses of the reviewed studies.
Thank you for this suggestions.
We add two tables to list the characteristics of the studies, as well as their strenghts and limitations
2. Additional figures and tables showing the features and performance of the specified studies being compared to each other.
Thank you for this suggestions.
We add a table to explain the characteristics and advantages of different AI approaches and some related examples
In the two added tables we explained the main features of the studies described in the text
3. What will be the possible studies in the future? It can be represented by roadmap specifying the years within 10 years to which the first AI technique 10 years ago was implemented for lung cancer up to the recent work.
We deepen the challenges section inserting possible solutions to current AI limitations.
We have added a final section about new perspectives and create an image showing all the possibilities provided by AI for lung cancer that will bring to a personalized medicine in the future
I upload the revised manuscript with a clean version first, followed by the old manuscript with marked revisions.
Thank you very much
Best regards
The authors
Reviewer 2 Report
Good work by the authors. But, it does not look like a comprehensive review paper. The authors should add some recent literature. Discuss advanced AI methods, such as contrastive learning, semi-supervised learning. They should also include some tables with the pros and cons of different models in order to better appreciate the contributions. The review paper should be way ahead of its time. They should look at some high impact review papers and than restructure the manuscript.
Author Response
The authors thank the reviewer for the work and precious suggestions.
We performed all the required changes as follows:
Good work by the authors. But, it does not look like a comprehensive review paper. The authors should add some recent literature. Discuss advanced AI methods, such as contrastive learning, semi-supervised learning. They should also include some tables with the pros and cons of different models in order to better appreciate the contributions. The review paper should be way ahead of its time. They should look at some high impact review papers and than restructure the manuscript.
Thank you very much for your comments.
We inserted a table showing the main features of different AI approaches, and tables showing the main characteristics of the studies cited in the manuscript, including advantages and limitations.
We extensively rewritten and modified the whole article, including new recently published studies in each section, we deepen the section of challenges, and added a section on new perspectives
I upload the revised manuscript with a clean version first, followed by the old manuscript with marked revisions.
Thank you very much
Best regards
The authors
Round 2
Reviewer 1 Report
The authors already revised their paper based on my comments.
Reviewer 2 Report
I think the paper is significantly improved and i recommend to accept this version with the condition of proof-read, i still found some errors and repetition in the manuscript.